# Adaptive Re-calibration Learning for Balanced Multimodal Intention Recognition

**Qu Yang**[1*]  **Xiyang Li**[1*]  **Fu Lin**[1†]  **Mang Ye**[1,2†]

[1] School of Computer Science, Wuhan University, Wuhan, China
[2] Taikang Center for Life and Medical Sciences, Wuhan University, Wuhan, China
{yangqu, lixiyang, linfu, yemang}@whu.edu.cn
https://github.com/yan9qu/NeurIPS25-ARL

## Abstract

Multimodal Intention Recognition (MIR) plays a critical role in applications such as intelligent assistants, service robots, and autonomous systems. However, in real-world scenarios, different modalities often vary significantly in informativeness, reliability, and noise levels. This leads to modality imbalance, where models tend to over-rely on dominant modalities, thereby limiting generalization and robustness. Although existing methods address this issue at either the sample or model level, they generally fail to account for its multi-level nature. To address this, we propose Adaptive Re-calibration Learning (ARL), a novel dual-path framework that models modality importance from both sample-wise and structural perspectives. ARL incorporates two key mechanisms: Contribution-Inverse Sample Calibration (CISC), which dynamically masks overly dominant modalities at the sample level to encourage attention to underutilized ones; and Weighted Encoder Calibration (WEC), which adjusts encoder weights based on global modality contributions to prevent overfitting. Experimental results on multiple MIR benchmarks demonstrate that ARL significantly outperforms existing methods in both accuracy and robustness, particularly under noisy or modality-degraded conditions.

## 1  Introduction

Understanding human intentions is paramount for creating truly intelligent and user-centric systems [1–6]. From predicting user needs in smart homes to enabling natural interactions with robots, the ability to accurately decipher what a person wants to achieve is crucial. Multimodal Intention Recognition (MIR) [7, 8] directly addresses this challenge by integrating information from diverse modalities, including visual cues, linguistic expressions, and auditory signals, to infer users' underlying goals. This capability is not just theoretical; it's the bedrock for advancing a wide range of real-world applications, including more intuitive intelligent assistants, sophisticated robotics, and safer, more responsive autonomous systems [9, 10].

However, achieving reliable intention recognition in multimodal settings is far from trivial [11, 12]. Real-world multimodal data is inherently heterogeneous: different modalities contribute unequally, suffer from varying levels of noise, and may be missing or degraded under certain conditions [13–15]. These challenges make effective integration of modalities both crucial and non-trivial, and they highlight the importance of addressing modality imbalance, which is a core factor that significantly affects MIR model robustness and generalization.

---

*Equal contribution.
†Corresponding author.

39th Conference on Neural Information Processing Systems (NeurIPS 2025).

Despite recent advances in MIR, modality imbalance remains a key challenge that limits model performance and generalization [16]. In realistic settings, different modalities vary significantly in terms of informativeness, reliability, and noise levels. This heterogeneity often causes models to over-rely on dominant modalities while underutilizing weaker but potentially critical signals. To address this, researchers have proposed various strategies. Early approaches such as gradient modulation [17] aim to balance modality contributions by globally adjusting learning rates during optimization. However, these methods typically lack fine-grained sensitivity to the varying importance of modalities at the sample level. More recently, Shapley value-based methods [14] have been introduced to more precisely estimate the marginal contributions of different modality combinations. While theoretically appealing, these methods suffer from exponential computational complexity with respect to the number of modalities, severely limiting their scalability in real-world MIR tasks. This limitation triggers a critical question: *How can we enable fine-grained, dynamic modeling of modality importance **without incurring prohibitive computational cost***?

Moreover, most existing methods focus solely on either sample-level weighting or model-level parameter adjustment [18–20], overlooking the multi-level nature of modality imbalance. In reality, imbalance occurs not only in how individual samples leverage modalities, but also in how the model itself encodes and integrates them structurally. Thus, methods that optimize only one level often fall short in handling the complexity of real-world multimodal scenarios. This observation leads to another fundamental question: *How can we jointly address modality imbalance at **both the input and architectural** levels to build more robust and generalizable MIR models?*

To this end, we propose Adaptive Re-calibration Learning (ARL), a novel framework that tackles modality imbalance through a dual-path calibration mechanism. ARL introduces two complementary strategies that work in tandem to address imbalance from both the input and model perspectives. The first strategy dynamically masks dominant modalities during training based on per-sample contribution estimates, encouraging the model to pay more attention to underrepresented modalities, a component referred to as Contribution-Inverse Sample Calibration (CISC). Concurrently, the second strategy adjusts the weights of modality-specific encoders according to their overall importance, which helps prevent overfitting to strong modalities and promotes balanced representation learning at the architectural level, known as Weighted Encoder Calibration (WEC). By integrating these two mechanisms, ARL achieves a more holistic and adaptive approach to modality balancing in MIR. Extensive experiments on multiple MIR benchmarks demonstrate that ARL outperforms state-of-the-art methods in both accuracy and robustness. Particularly in noisy or modality-deprived settings, ARL shows superior generalization and resilience, highlighting its practical value in real-world deployment. Our contributions are threefold:

- We propose Adaptive Re-calibration Learning (ARL), a unified framework that addresses modality imbalance in Multimodal Intention Recognition by jointly calibrating modality importance at both the input and architectural levels.

- We design two complementary components within ARL: Contribution-Inverse Sample Calibration (CISC), which dynamically down-weights dominant modalities at the sample level, and Weighted Encoder Calibration (WEC), which adjusts encoder weights based on modality importance, enabling balanced and adaptive multimodal representation learning.

- We conduct extensive experiments on multiple MIR benchmarks, demonstrating that ARL consistently outperforms state-of-the-art methods in both accuracy and robustness, especially under noisy or modality-degraded conditions.

## 2 Related Work

### 2.1 Multimodal Intention Understanding

Multimodal intention recognition (MIR) seeks to infer user intent by integrating diverse signals from multiple modalities, such as text, audio, and visual information [21–24]. Early unimodal methods, including text-based intent classification [25] and acoustic emotion recognition [26], often simplify the inherent complexity of human intention by relying on a single source of data. In contrast, multimodal learning approaches provide a more comprehensive understanding by leveraging the complementary nature of different modalities. Representative methods include MulT [27], which models inter-modal interactions through pairwise attention mechanisms, and MISA [28], which captures both shared and

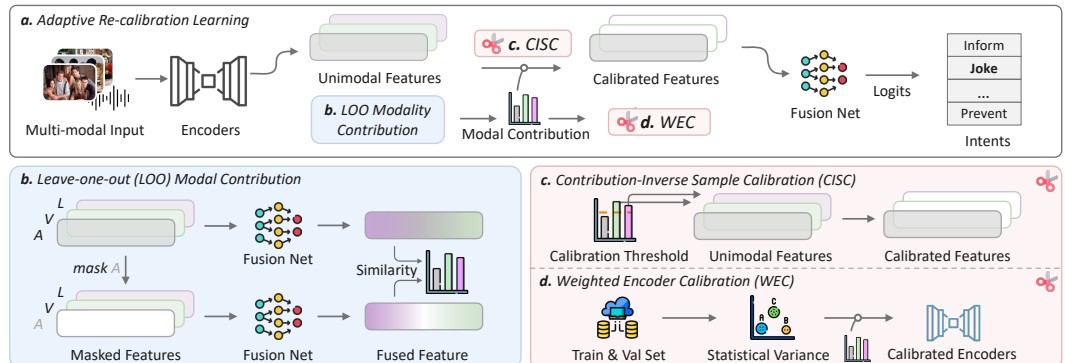

Figure 1: The architecture of Adaptive Re-calibration Learning (ARL). ARL integrates three components: (b) Leave-One-Out (LOO) Modality Contribution, (c) Contribution-Inverse Sample Calibration (CISC), and (d) Weighted Encoder Calibration (WEC). LOO assesses modality importance by masking inputs and measuring similarity. CISC refines features using a contribution threshold. WEC adjusts encoders with consistency gap from training and validation sets. The process concludes with Fusion Net for intent prediction.

modality-specific representations to enhance generalization. MAG-BERT [29] further extends this line of research by integrating nonverbal modalities into the BERT architecture using a multimodal adaptation gate, thereby enabling the model to incorporate contextual signals beyond text.

More recently, researchers have explored advanced strategies to further improve multimodal learning in MIR. Contrastive learning methods [30] have been proposed to better align and distinguish representations across modalities by promoting consistency while preserving unique modality characteristics. In addition, modality-aware prompting techniques [11] have been introduced to dynamically guide model behavior based on the specific combination of available modalities. Despite these advancements, one of the major challenges that remains is the issue of sample-specific modality imbalance. In many real-world scenarios, certain modalities may be noisy, incomplete, or less informative for particular samples, which can negatively impact model performance. Addressing this challenge is essential for developing robust and adaptable multimodal intention recognition systems.

## 2.2 Imbalanced multimodal learning

Imbalanced multimodal learning arises from inherent variations in informativeness and reliability across modalities, often leading models to overemphasize dominant signals while neglecting weaker, yet potentially crucial, complementary cues [31–33]. Traditional techniques, such as loss re-weighting and modality-specific augmentation, attempt to balance modality influence but often lack adaptability to sample-specific variations [34–37]. Gradient modulation [17] represents a more dynamic approach, adjusting training updates to mitigate the dominance of stronger modalities. However, these methods typically operate at a dataset level and lack the instance-level adaptivity essential, where modality importance is context-dependent.

Modality dropout [38–40] and contribution-aware learning [41, 42] are prominent strategies for managing modality imbalance. Modality dropout enhances robustness by randomly masking features from dominant modalities during training, encouraging reliance on weaker signals. However, its heuristic nature can inadvertently suppress critical information. Contribution-aware learning offers a more principled approach, employing techniques like Shapley values [14] to estimate and balance modality contributions. While theoretically sound, the computational complexity of Shapley value calculation hinders its practicality in real-time MIR applications. Thus, achieving an optimal balance between adaptivity, computational efficiency, and predictive accuracy remains a significant challenge in imbalanced multimodal learning.

## 3 Methodology

The **Adaptive Re-calibration Learning** (ARL) framework, shown in Fig. 1, addresses the modality imbalance problem in Multimodal Intention Recognition (MIR) by integrating three key components:

**Leave-One-Out** (LOO) modality contribution estimation, **Contribution-Inverse Sample Calibration** (CISC), and **Weighted Encoder Calibration** (WEC). As noted in Sec. 1, modality imbalance, where certain modalities dominate due to differences in informativeness and reliability, poses a significant challenge for MIR. ARL mitigates this through an efficient and adaptive strategy. It starts with unimodal encoders extracting features from language, visual, and acoustic inputs via pre-trained models. The LOO module then evaluates each modality's contribution to the fused representation. Based on these estimates, CISC calibrates features at the sample level by masking dominant modalities to encourage the use of weaker ones. In parallel, WEC refines the modality-specific encoders by considering both their contribution and stability during training, promoting more balanced learning. A complete list of variables and their definitions can be found in Appendix A.1.

### 3.1 Problem Formulation

To provide a clear foundation for ARL, we first present a general framework for multimodal intention recognition. Given a source input $m = \{L, V, A\}$, where $L$, $V$, and $A$ represent the language, visual, and acoustic modalities, respectively, the goal is to fuse these modalities into a unified representation for intention recognition. Each modality is encoded into its respective feature representations:

$$m_t = \text{TextEmbed}(L) \in \mathbb{R}^{l_t \times d_t}, \tag{1}$$

$$m_a = \text{AudioEmbed}(A) \in \mathbb{R}^{l_a \times d_a}, \tag{2}$$

$$m_v = \text{VideoEmbed}(V) \in \mathbb{R}^{l_v \times d_v}, \tag{3}$$

where $m_t$, $m_a$ and $m_v$ are the extracted feature representations of each modality. Here, $l_{t/a/v}$ denotes sequence lengths (text tokens/audio frames/video frames), and $d_{t/a/v}$ represents feature dimensions for respective modalities. After obtaining the shallow features for each modality, a critical step is to temporally align these heterogeneous sequences to enable cross-modal interaction. This alignment process maps the features of each modality into a shared space while synchronizing their temporal dynamics. Formally, given the modality-specific features $m_t, m_a, m_v$, the alignment operation can be expressed as:

$$h_t, h_a, h_v = \text{SeqAligned}(m_t, m_a, m_v) \in \mathbb{R}^{l \times d}, \tag{4}$$

where $h_t$, $h_a$ and $h_v$ are the aligned feature representations of each modality. Here, $l$ indicates the unified sequence length after alignment, and $d$ is the projected feature dimension. Once the high-level, aligned features are obtained, they are fused through multimodal fusion to obtain the final representation for intent prediction:

$$h_{\text{m}} = \mathcal{F}(h_t, h_a, h_v) \in \mathbb{R}^d, \tag{5}$$

where $\mathcal{F}(\cdot)$ denotes the multimodal fusion function. The final intent prediction is made by applying a softmax function to the fused representation:

$$logits = \text{Softmax}(W^T h_{\text{m}} + b), \tag{6}$$

with $W \in \mathbb{R}^{d \times C}$ and $b \in \mathbb{R}^C$ being trainable parameters, where $C$ denotes the number of categories. The whole framework preserves modality-specific patterns while enabling cross-modal interaction through shared alignment. To formally quantify the issue of modality imbalance, we introduce the Modality Imbalance Index (MII), a metric derived from our LOO contributions. A detailed definition and formulation are provided in Appendix A.3.

### 3.2 Leave-One-Out Modality Valuation

Quantifying the contribution of each modality (language, visual, and acoustic) is fundamental to tackling modality imbalance in multi-modal intent recognition (MIR) systems. The Leave-One-Out (LOO) method offers a practical and efficient solution for this, contrasting with the computationally demanding Shapley value approach, which scales exponentially with the number of modalities.

For a given sample $i$, we start with features from the language ($m_t^i$), acoustic ($m_a^i$), and visual ($m_v^i$)) modalities. These are combined to produce a fused representation that encapsulates the joint information:

$$\mathbf{F}_i = \text{Fuse}(\text{SeqAligned}(m_t^i, m_a^i, m_v^i)), \tag{7}$$

where *Fuse($\cdot$) denotes the fusion function provided by the base multimodal learning architecture.* Importantly, ARL operates as a lightweight plug-in module and does not modify or participate in the modality fusion process. To evaluate the importance of a specific modality $m \in \{t, a, v\}$, we mask that modality by replacing its features with a zero vector. This simulates the absence of $m$ and allows us to compute a modified fused representation:

$$\mathbf{F}_i^{(-m)} = \text{Fuse}(\text{SeqAligned}(\tilde{m}_t^i, \tilde{m}_a^i, \tilde{m}_v^i)), \tag{8}$$

where $\tilde{m}_t^i$, $\tilde{m}_a^i$, and $\tilde{m}_v^i$ denote the inputs with modality $m$ masked (e.g., if $m = a$, then $\tilde{m}_a^i = \mathbf{0}$, while $\tilde{m}_t^i = m_t^i$ and $\tilde{m}_v^i = m_v^i$). $\mathbf{F}_i^{(-m)}$ represents the fused representation obtained when the modality $m$ is masked, allowing us to analyze the impact of $m$ on the overall representation.

The contribution of modality $m$, denoted $\delta_i^{(m)}$, is determined by comparing the original fused representation $\mathbf{F}_i$ with the modified one $\mathbf{F}_i^{(-m)}$ using cosine similarity. This metric measures the angular difference between the two vectors, indicating the impact of modality $m$ on the fused output. A high cosine similarity (close to 1) means the representations are nearly aligned, suggesting that masking $m$ has minimal impact and thus $m$ contributes little. A low similarity (closer to 0 or negative) indicates a significant shift, implying that $m$ is highly influential. The contribution is formalized as:

$$\delta_i^{(m)} = \frac{\exp(-\eta \cdot \text{Sim}(\mathbf{F}_i, \mathbf{F}_i^{(-m)}))}{\sum_{m'} \exp(-\eta \cdot \text{Sim}(\mathbf{F}_i, \mathbf{F}_i^{(-m')}))}, \tag{9}$$

where $\text{Sim}(\cdot)$ is the cosine similarity, and $\eta$ is a sensitivity parameter that adjusts the sharpness of the contribution scores. A larger $\eta$ amplifies differences in similarity scores, making the distinction between high- and low-contributing modalities more pronounced.

Unlike Shapley methods requiring $2^M$ combinations (where $M$ is the number of modalities), LOO needs only $M$ forward passes per sample, reducing complexity from exponential to linear—crucial for real-time MIR. This process, depicted in Fig. 1 (b), enables sample-specific modality valuation, allowing the system to adapt dynamically to varying modality importance across different inputs.

### 3.3 Contribution-Inverse Sample Calibration

Building on the modality contributions derived from LOO, Contribution-Inverse Sample Calibration (CISC) addresses modality dominance by adjusting features at the sample level. The principle is to reduce reliance on overly influential modalities, encouraging the model to utilize weaker but potentially valuable ones, thus promoting a balanced multi-modal representation.

For each sample $i$ and modality $m$, we use the LOO contribution $\delta_i^{(m)}$ and compare it to a threshold $\tau$. If $\delta_i^{(m)} \geq \tau$, indicating that modality $m$ dominates the fused representation, we mask its features by setting them to zero. Otherwise, the features remain intact:

$$\tilde{x}_i^{(m)} = \begin{cases} x_i^{(m)}, & \text{if } \delta_i^{(m)} < \tau, \\ \mathbf{0}, & \text{if } \delta_i^{(m)} \geq \tau, \end{cases} \tag{10}$$

where $x_i^{(m)}$ are the original features, and $\tilde{x}_i^{(m)}$ are the calibrated features. By removing the influence of a dominant modality (e.g., audio in a noisy environment), the model must rely on the remaining modalities, enhancing robustness (see Fig. 1 (c)). The threshold $\tau$ governs the masking aggressiveness; a lower $\tau$ masks more modalities, potentially improving balance but risking information loss, while a higher $\tau$ is more conservative. The calibrated features are then fused:

$$\tilde{F}_i = \text{Fuse}(\text{SeqAligned}(\tilde{x}_i^{(t)}, \tilde{x}_i^{(a)}, \tilde{x}_i^{(v)})), \tag{11}$$

driving intent prediction. This per-sample adjustment adapts to context-specific modality importance, making the MIR system versatile across diverse scenarios.

Our use of hard masking (zeroing out features) is a deliberate design choice. The motivation is to introduce a strong regularization effect, decisively breaking the model's over-reliance on dominant modalities. An alternative could be soft masking, where features are only attenuated. We empirically validate our choice in Sec. 4.2, where we show that hard masking leads to a more robust and balanced representation.

### 3.4 Weighted Encoder Calibration

While CISC handles sample-level imbalances, Weighted Encoder Calibration (WEC) ensures long-term stability and balance by adjusting modality-specific encoders based on their global performance. We introduce the **purity score** as a measure of class consistency within clusters. It quantifies the proportion of the dominant true class in each cluster and is formally defined as:

$$\text{Purity}(y, c) = \frac{1}{n} \sum_{k=1}^{K} \max_{j \in \mathcal{Y}} \#\{i : y_i = j, \, c_i = k\}, \tag{12}$$

where $y \in \mathbb{N}^n$ denotes the vector of ground-truth class labels, $c \in \mathbb{N}^n$ represents the cluster assignments, $K$ is the number of clusters, $\mathcal{Y}$ is the set of all class labels, and $\#\{i : y_i = j, \, c_i = k\}$ counts the number of samples from class $j$ in cluster $k$. This metric reflects how well the clustering preserves the underlying class structure. Using this metric, we assess the stability of each modality across training and validation sets based on the consistency gap:

$$\mathcal{P}_m = |p_m^{\text{train}} - p_m^{\text{val}}|, \tag{13}$$

where $p_m^{\text{train}}$ and $p_m^{\text{val}}$ are purity scores for modality $m$ on training and validation sets. A high $\mathcal{P}_m$ indicates inconsistency, such as overfitting.

Combining this with the average LOO contribution $\delta^{(m)}$, WEC computes a dynamic weight:

$$w_m = \tanh(\lambda \cdot \mathcal{P}_m + \alpha \cdot \delta^{(m)}), \tag{14}$$

where $\lambda$ and $\alpha$ tune the influence of stability and contribution. The encoder parameters are then adjusted:

$$\theta_{\text{new}}^{(m)} = w_m \cdot \theta_{\text{init}}^{(m)} + (1 - w_m) \cdot \theta_{\text{current}}^{(m)}, \tag{15}$$

shifting unstable or underused encoders toward their initial state while preserving learned parameters for stable ones. Applied every $T$ epochs (see Fig. 2), WEC prevents long-term bias, enhancing generalization across training.

### 3.5 The Synergy of CISC and WEC

CISC and WEC synergistically mitigate modality imbalance through complementary short-term and long-term calibration. CISC adjusts features per batch, masking dominant modalities to emphasize weaker ones, adapting to sample-specific variations (Fig. 1(c)). WEC, invoked periodically, refines encoders using aggregated contribution and stability metrics, ensuring balanced learning over time. During training, CISC operates at each batch, followed by WEC every $T$ epochs, creating an iterative optimization loop. This dual-phase strategy, visualized across Fig. 1, enhances ARL's ability to handle dynamic modality importance, delivering superior accuracy and efficiency, vital for real-world MIR applications like robotics and smart assistants. To provide a clear, step-by-step overview of the entire training process, we include a detailed pseudocode of the ARL training procedure in Appendix A.4

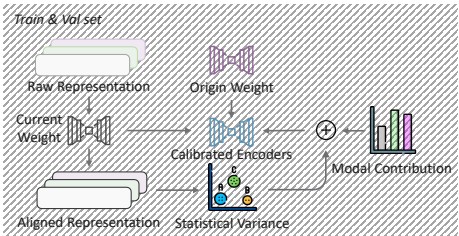

Figure 2: Overview of WEC. The module processes training and validation sets to compute consistency gap and modal contributions, adjusting encoder to ensure balance across modalities.

## 4 Experiments

**Datasets**. We evaluate our ARL framework on two multimodal benchmarks: MIntRec [21] for intent recognition, and MOSI [43] for sentiment analysis. In addition to multimodal intention recognition, we posit that multimodal sentiment analysis also suffers from modal imbalance issues. Therefore, we incorporate MOSI dataset as additional evaluation benchmark. The MIntRec dataset comprises 2,224 samples, split into 1,334 for training, 445 for validation, and 445 for testing. It supports two

Table 1: Performance comparison of various multimodal intention understanding methods on the MIntRec dataset. Our proposed Adaptive Re-calibration Learning (ARL) framework consistently improves the performance of baseline methods (MAG_BERT, MulT, and TCL_MAP) across all metrics for both twenty-class fine-grained classification and binary coarse-grained classification tasks.

| Method | Twenty-class | | | | Binary | | | |
|---|---|---|---|---|---|---|---|---|
| | ACC | F1(WF1) | P(WP) | R | ACC | F1(WF1) | P(WP) | R |
| MAG_BERT [29] | 72.65 | 68.64 | 69.08 | 69.28 | 89.24 | 89.1 | 89.1 | 89.13 |
| w/ ARL | **74.38** | **71.66** | **71.55** | **72.89** | **90.11** | **90.01** | **89.92** | **90.14** |
| Δ *Improvement* | +1.73 | +3.02 | +2.47 | +3.61 | +0.87 | +0.91 | +0.82 | +1.01 |
| MulT [27] | 72.52 | 69.25 | 70.25 | 69.24 | 89.19 | 89.07 | 89.02 | 89.18 |
| w/ ARL | **73.71** | **70.82** | **72.06** | **70.45** | **89.89** | **89.78** | **89.70** | **89.88** |
| Δ *Improvement* | +1.19 | +1.57 | +1.81 | +1.21 | +0.70 | +0.71 | +0.68 | +0.70 |
| TCL_MAP [11] | 73.62 | 73.31 | 73.72 | 70.50 | 89.66 | 89.69 | 89.84 | 89.84 |
| w/ ARL | **75.28** | **75.16** | **75.71** | **73.35** | **90.56** | **90.57** | **90.59** | **90.54** |
| Δ *Improvement* | +1.66 | +1.85 | +2.00 | +2.85 | +0.90 | +0.88 | +0.75 | +0.70 |

Table 2: Comparative experimental results of the sentiment analysis models on the public MOSI dataset. The best results are highlighted in **bold**.

| Method | ACC | F1(WF1) | P(WP) | R |
|---|---|---|---|---|
| MAG_BERT | 42.35 | 36.54 | 37.51 | 37.36 |
| w/ ARL | **43.22** | **37.44** | **38.23** | **38.53** |
| Δ *Improv.* | +0.87 | +0.90 | +0.72 | +1.17 |
| MulT | 42.42 | 37.24 | 37.00 | 39.05 |
| w/ ARL | **43.44** | **38.50** | **38.97** | **40.42** |
| Δ *Improv.* | +1.02 | +1.26 | +1.97 | +1.37 |
| TCL_MAP | 42.13 | 40.54 | 42.68 | 37.17 |
| w/ ARL | **43.73** | **42.60** | **43.73** | **38.38** |
| Δ *Improv.* | +1.60 | +2.06 | +1.05 | +1.21 |

Table 3: Impact of hyperparameter $\alpha$ on the MOSI. Best results per metric are in **bold**.

| $\alpha$ | ACC | F1 | P | R | AVG |
|---|---|---|---|---|---|
| 0.01 | 42.42 | 36.32 | 36.77 | 38.23 | 38.43 |
| 0.1 | 42.56 | 35.75 | 37.58 | 37.35 | 38.31 |
| 0.5 | 42.78 | 37.40 | 38.21 | 38.07 | 39.11 |
| 1 | **43.80** | 36.69 | 37.80 | 37.91 | 39.05 |
| 5 | 43.22 | **37.44** | 38.23 | **38.53** | **39.35** |
| 10 | 41.62 | 37.30 | **38.32** | 37.98 | 38.80 |

levels of intent classification: coarse-grained, with binary labels distinguishing between expressing emotions and achieving goals, and fine-grained, with twenty labels (11 for expressing emotions and 9 for achieving goals). The MOSI dataset consists of 2,199 samples, divided into 1,284 for training, 229 for validation, and 686 for testing, with sentiment scores ranging from -3 (highly negative) to 3 (highly positive). For MOSI, we discretize the continuous sentiment scores into labels from 0 to 6 for classification tasks.

**Evaluation Metrics**. For MIntRec, we report Accuracy (ACC), F1-score (F1), Precision (P), and Recall (R) to evaluate fine-grained intention recognition performance. For MOSI, we apply the same metrics to assess the 7-class sentiment classification task. Additionally, when integrating with TCL-MAP [11], which uses weighted F1 (WF1) and weighted Precision (WP), we include these metrics to ensure compatibility with its evaluation standards.

**Implementation Details**. Our plug-and-play module is incorporated into three established methods: MAG-BERT [29], MulT [27], and TCL-MAP [11]. We adopt hyper-parameters such as learning rate, batch size, and optimizer settings from the publicly released configurations of these baseline methods. We tune specific hyper-parameters for ARL, including masking threshold in CISC, weight adjustment factor in WEC, optimizing them via grid search. For MIntRec, we use pre-extracted features with dimensions 768 for text, 256 for visual, and 768 for acoustic. For MOSI, feature dimensions are 768 for text, 20 for video, and 5 for audio. All experiments are conducted on 4 NVIDIA 4090 GPUs.

Table 4: Ablation study of the proposed model on MIntRec. The best results are highlighted in **bold**.

| Method | CISC | WEC | Twenty-class | | | | | Binary | | | | |
|---|---|---|---|---|---|---|---|---|---|---|---|---|
| | | | ACC | F1/WF1 | P/WP | R | AVG | ACC | F1/WF1 | P/WP | R | AVG |
| MAG_BERT | | | 72.65 | 68.64 | 69.08 | 69.28 | 69.91 | 89.24 | 89.10 | 89.10 | 89.13 | 89.14 |
| | ✓ | | 72.81 | 68.81 | 69.79 | 69.32 | 70.18 | 89.55 | 89.43 | 89.41 | 89.55 | 89.48 |
| | | ✓ | 72.92 | 69.58 | 69.94 | 70.34 | 70.69 | 89.66 | 89.57 | 89.46 | 89.79 | 89.62 |
| | ✓ | ✓ | **74.38** | **71.66** | **71.55** | **72.89** | **72.62** | **90.11** | **90.01** | **89.92** | **90.14** | **90.04** |
| MulT | | | 72.52 | 69.25 | 70.25 | 69.24 | 70.31 | 89.19 | 89.07 | 89.02 | 89.18 | 89.11 |
| | ✓ | | 73.03 | 69.80 | 71.04 | 69.16 | 70.75 | 89.44 | 89.33 | 89.23 | 89.48 | 89.37 |
| | | ✓ | 72.80 | 69.74 | 70.62 | 70.00 | 70.79 | 89.21 | 89.05 | 89.14 | 88.97 | 89.09 |
| | ✓ | ✓ | **73.71** | **70.82** | **72.06** | **70.45** | **71.76** | **89.89** | **89.78** | **89.70** | **89.88** | **89.81** |
| TCL_MAP | | | 73.62 | 73.31 | 73.72 | 70.50 | 72.78 | 89.66 | 89.69 | 89.84 | 89.84 | 89.75 |
| | ✓ | | 73.93 | 73.76 | 74.37 | 71.50 | 73.39 | 90.34 | 90.30 | 90.38 | 89.92 | 90.23 |
| | | ✓ | 74.16 | 73.37 | 73.30 | 71.01 | 72.96 | 90.00 | 90.02 | 90.11 | 90.08 | 90.05 |
| | ✓ | ✓ | **75.28** | **75.16** | **75.71** | **73.35** | **74.87** | **90.56** | **90.57** | **90.59** | **90.54** | **90.56** |

## 4.1 Comparison with the State-of-the-art

As shown in Tabs. 1 and 2, our experimental analysis on the MIntRec and CMU-MOSI datasets demonstrates the consistent effectiveness of the ARL framework in enhancing multimodal intention recognition by addressing modality imbalance. On MIntRec, we evaluated state-of-the-art methods such as MAG-BERT, MulT, and TCL-MAP, and found that integrating ARL improves performance in both twenty-class and binary classification tasks. For example, TCL-MAP combined with ARL achieves a twenty-class accuracy of 75.28 and a binary accuracy of 90.56, with consistent gains across accuracy, F1 score, precision, and recall. Similarly, on CMU-MOSI, a benchmark for multimodal sentiment analysis, ARL improves all baseline methods. TCL-MAP, in particular, shows the most notable recalibration of modality importance, reaching an accuracy of 43.73. These results highlight that ARL not only improves performance but also adaptively recalibrates modality contributions to address dataset-specific challenges, making it a versatile framework for advancing multimodal learning. To further contextualize the performance of ARL, we conducted additional experiments comparing it directly against representative imbalanced learning methods and evaluating its plug-and-play capability on other recent state-of-the-art models for multimodal understanding. In all cases, ARL demonstrated significant improvements, validating its effectiveness and versatility. The detailed results of these comprehensive comparisons are provided in Appendix A.5.

## 4.2 Ablation Studies

**Effectiveness of Proposed Components.** To assess the contributions of the core components in the ARL framework, namely CISC and WEC, we conduct ablation studies on the MIntRec dataset for both twenty-class and binary classification tasks. As shown in Tab. 4, CISC and WEC each improve baseline performance when applied independently. CISC reduces modality bias by masking dominant modalities, while WEC refines modality-specific encoders based on purity variation and LOO contribution. Their combination yields a synergistic effect: in the twenty-class task, MAG_BERT's AVG increases from 69.91 to 72.62, and TCL_MAP reaches 74.87. For binary classification, despite already strong baselines, ARL consistently boosts performance, with TCL_MAP improving to 90.56. Although WEC alone slightly reduces MulT's binary AVG, its combination with CISC recovers and enhances performance. These results highlight ARL's adaptability and its effectiveness in addressing modality imbalance through complementary mechanisms.

**Sensitivity Analysis of Calibration Weight $\alpha$ and Masking Threshold $\tau$.** To evaluate the role of modality contributions estimated by the LOO method in adjusting calibration weights within WEC, we conducted a series of experiments on the MOSI dataset. The results, summarized in Tab. 3, focus

Table 5: Sensitivity analysis of the CISC masking threshold $\tau$ on the MIntRec (twenty-class) dataset using the MulT model. The baseline performance corresponds to $\tau = 1.0$. Best results are in bold.

| $\tau$ | ACC | F1 | P | R |
|---|---|---|---|---|
| 1.0 | 72.52 | 69.25 | 70.25 | 69.24 |
| 0.9 | 72.58 | 69.15 | 70.04 | 69.15 |
| 0.7 | 71.24 | 67.98 | 68.70 | 68.11 |
| 0.5 | **73.71** | **70.82** | **72.06** | **70.45** |
| 0.3 | 73.26 | 69.94 | 71.23 | 69.64 |
| 0.1 | 70.79 | 67.16 | 68.27 | 66.52 |

Table 6: Comparison of Hard Masking (our ARL) and Soft Masking on MIntRec with MAG-BERT. The baseline performance corresponds to not using a mask. Best results are in bold.

| **Method** | ACC | F1 | P | R |
|---|---|---|---|---|
| Baseline | 72.65 | 68.64 | 69.08 | 69.28 |
| w/ HardMask | **74.38** | **71.66** | **71.55** | **72.89** |
| $\Delta$ *Improv.* | +1.73 | +3.02 | +2.47 | +3.61 |
| w/ SoftMask | 73.24 | 67.31 | 70.14 | 69.57 |
| $\Delta$ *Improv.* | +0.59 | -1.33 | +1.06 | +0.71 |

on the effect of different $\alpha$ values (0.01, 0.1, 0.5, 1, 5, 10) on the performance of the MAG_BERT model. Here, $\alpha$ controls the influence of LOO-derived contributions on WEC. The results show that when $\alpha = 1$, the model achieves the highest accuracy (43.80) and a relatively high recall (37.91), although the F1 score (36.69) is slightly lower than that at $\alpha = 5$. At $\alpha = 5$, the model obtains the highest F1 score (37.44) and peak recall (38.53), resulting in the best overall performance (AVG = 39.35). In contrast, setting $\alpha$ too low (e.g., 0.01) or too high (e.g., 10) leads to performance degradation across all metrics, indicating the importance of selecting an appropriate $\alpha$ value to balance performance, $\alpha = 1$ and $\alpha = 5$ yield the best results, confirming that incorporating LOO-based modality contributions into WEC can effectively enhance model performance when properly calibrated.

To further demonstrate the robustness of our ARL framework, we provide a sensitivity analysis for the Contribution-Inverse Sample Calibration (CISC) masking threshold, $\tau$. This hyperparameter governs the aggressiveness of the masking mechanism; a lower $\tau$ means modalities are masked more frequently. The analysis was conducted using the MulT model on the MIntRec (twenty-class) dataset.

As shown in Tab. 5, the model achieves optimal performance across all metrics when $\tau = 0.5$. A value of 1.0 is equivalent to the baseline model without CISC. While performance degrades with very aggressive masking (e.g., $\tau = 0.1$) or overly conservative masking, there is a stable range where the model benefits from the calibration. We observed similar trends across our other experimental setups. This analysis demonstrates that while the hyperparameter is important, its optimal value can be reliably identified via standard tuning procedures.

**Effectiveness of Hard Masking** To validate our choice of hard masking in the CISC module, we conducted an experiment comparing its performance against a soft masking alternative. In the soft masking setup, instead of being zeroed out, the features of a dominant modality are attenuated (e.g., multiplied by a factor of $1 - \delta_i^{(m)}$).

The results, presented in Tab. 6, clearly show that our proposed hard masking approach yields significant and consistent improvements across all evaluation metrics. While soft masking provided a modest improvement in Accuracy and Precision, it was detrimental to the F1-score and Recall. We hypothesize that this is because an attenuated signal is insufficient to compel the model to fundamentally alter its strategy. Hard masking, in contrast, forces the model to actively learn from underutilized modalities, leading to superior F1 and Recall scores which indicate a more robust and balanced model.

**Generalizability under Missing Modalities.** To assess the robustness of the ARL framework, we conduct ablation studies under missing modality conditions on the CMU-MOSI dataset. We test three modality pairs: $\{a, v\}$ (audio-visual), $\{v, t\}$ (visual-text), and $\{a, t\}$ (audio-text), evaluating performance improvements for both MAG-BERT and MulT models. As shown in Tab. 7, ARL consistently enhances accuracy ($ACC_7$, $ACC_2$) and F1 scores ($F1_7$, $F1_2$) across all settings. For example, in the $\{a, v\}$ scenario, ARL increases MulT's $ACC_7$ from 22.13 to 23.09 and $ACC_2$ from 59.79 to 60.67. The largest gains occur in the $\{a, t\}$ setting, where MAG-BERT's $ACC_7$ rises by 1.57 (to 42.82) and $ACC_2$ by 1.08 (to 81.98), underscoring ARL's ability to leverage textual information, which is vital for sentiment analysis. These results highlight ARL's adaptability and effectiveness in mitigating modality imbalance across diverse configurations and architectures.

Table 7: Performance under missing modality scenarios on CMU-MOSI. Best results are in **bold**.

| Method | {a,v} | | | | {v,t} | | | | {a,t} | | | |
|---|---|---|---|---|---|---|---|---|---|---|---|---|
| | $ACC_7$ | $F1_7$ | $ACC_2$ | $F1_2$ | $ACC_7$ | $F1_7$ | $ACC_2$ | $F1_2$ | $ACC_7$ | $F1_7$ | $ACC_2$ | $F1_2$ |
| MAG_BERT | 17.78 | 7.50 | 48.13 | 45.29 | 40.00 | 35.06 | 81.07 | 80.79 | 41.25 | 35.69 | 80.90 | 80.56 |
| w/ ARL | **19.30** | **8.24** | **48.80** | **46.77** | **41.02** | **35.89** | **81.63** | **81.43** | **42.82** | **35.70** | **81.98** | **81.73** |
| $\Delta$ Improvement | +1.52 | +0.74 | +0.67 | +1.48 | +1.02 | +0.83 | +0.56 | +0.64 | +1.57 | +0.01 | +1.08 | +1.17 |
| MulT | 22.13 | 15.93 | 59.79 | 58.74 | 40.90 | 36.57 | 81.24 | 81.03 | 41.19 | 36.55 | 81.22 | 80.88 |
| w/ ARL | **23.09** | **16.22** | **60.67** | **60.21** | **41.89** | **37.03** | **81.54** | **81.19** | **42.27** | **36.76** | **81.51** | **81.30** |
| $\Delta$ Improvement | +0.96 | +0.29 | +0.88 | +1.47 | +0.99 | +0.46 | +0.30 | +0.16 | +1.08 | +0.21 | +0.29 | +0.42 |

**Computational Cost and Scalability of LOO.**
To evaluate the practicality of our Leave-One-Out (LOO) modality valuation method compared to Shapley-based approaches, we analyze computational cost with respect to two key factors: feature dimension and number of modalities. This addresses scalability concerns raised in Sec. 1. As shown in Fig. 3, both methods scale linearly with feature dimension, as expected. However, they differ significantly in scalability with respect to the number of modalities. Shapley-based methods suffer from exponential complexity ($O(n2^n)$), due to evaluating all possible modality subsets. In contrast, LOO demonstrates linear growth ($O(n)$), requiring only a single forward pass per modality. This makes LOO far more scalable and suitable for real-world MIR applications with multiple modalities.

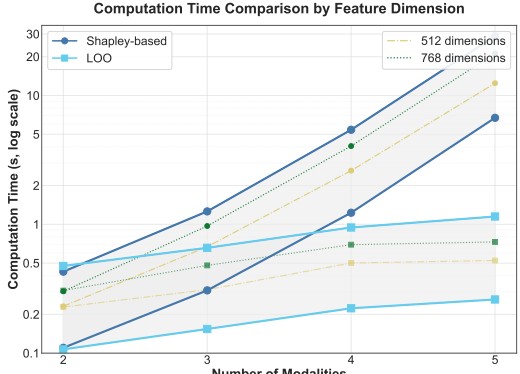

Figure 3: Computational cost for LOO and Shapley methods.

## 5   Conclusion and Discussion

In conclusion, we have presented Adaptive Re-calibration Learning (ARL), a novel framework designed to mitigate modality imbalance in Multimodal Intention Recognition and enhance the robustness of multimodal learning. ARL leverages our proposed Leave-One-Out (LOO) modality valuation method, a computationally efficient alternative to Shapley values, to power a two-phase calibration strategy. Through Contribution-Inverse Sample Calibration (CISC) and Weighted Encoder Calibration (WEC), ARL dynamically refines model learning by addressing both sample-specific and modality-level imbalances. Our extensive evaluations across MIntRec and MOSI datasets demonstrate ARL's effectiveness in boosting performance over strong baselines. Crucially, ARL overcomes the scalability bottleneck inherent in Shapley-based approaches, offering a practical and efficient solution for real-world MIR applications.

**Limitations of ARL.** Although ARL has achieved remarkable results in alleviating the modality imbalance problem in multimodal intent recognition, it still has some limitations. First, the performance of ARL depends on the expressiveness of the underlying model and the design of the fusion mechanism. If the base model itself has biases in understanding certain modalities or lacks sufficient feature extraction capabilities, ARL may not be able to fully compensate for these shortcomings. In addition, the computational overhead increases. Although the LOO (Leave-One-Out) method has linear complexity compared to Shapley values, in large-scale datasets or multimodal tasks, each forward pass used to assess modality contributions still introduces extra computational costs. The dual calibration mechanisms of WEC and CISC improve performance, but they also lead to increased training time and resource consumption.

**Acknowledgements.** This work is partially supported by the National Key Research and Development Program of China (2024YFC3308400), National Natural Science Foundation of China under Grant (62176188, 62361166629), Major Project of Science and Technology Innovation of Hubei Province (2024BCA003, 2025BEA002).

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

# A   Technical Appendices and Supplementary Material

## A.1   Variable summary

See Tab. 8

Table 8: Variables and Definitions in the ARL

| Variable | Definition |
|---|---|
| $m = \{L, V, A\}$ | Input source modalities: Language (L), Visual (V), and Acoustic (A). |
| $m_t$ | Text modality feature: $m_t = \text{TextEmbed}(L) \in \mathbb{R}^{l_t \times d_t}$. |
| $m_a$ | Audio modality feature: $m_a = \text{AudioEmbed}(A) \in \mathbb{R}^{l_a \times d_a}$. |
| $m_v$ | Video modality feature: $m_v = \text{VideoEmbed}(V) \in \mathbb{R}^{l_v \times d_v}$. |
| $l_{t/a/v}$ | Sequence lengths of respective modalities (e.g., tokens, frames). |
| $d_{t/a/v}$ | Feature dimensions of each modality. |
| $h_t, h_a, h_v$ | Aligned multimodal features: $\text{SeqAligned}(m_t, m_a, m_v) \in \mathbb{R}^{l \times d}$. |
| $l$ | Unified sequence length after alignment. |
| $d$ | Projected feature dimension. |
| $h_m$ | Final fused representation: $h_m = \mathcal{F}(h_t, h_a, h_v) \in \mathbb{R}^d$. |
| $\mathcal{F}(\cdot)$ | Multimodal fusion function. |
| logits | Intent prediction output: $\text{Softmax}(W^T h_m + b)$. |
| $W$ | Weight matrix: $W \in \mathbb{R}^{d \times C}$. |
| $b$ | Bias vector: $b \in \mathbb{R}^C$. |
| $C$ | Number of classes. |
| $\mathbf{F}_i$ | Original fused representation for sample $i$: $\text{Fuse}(\text{SeqAligned}(m_t^i, m_a^i, m_v^i))$. |
| $\mathbf{F}_i^{(-m)}$ | Modified fused representation with modality $m$ masked: $\text{Fuse}(\text{SeqAligned}(\tilde{m}_t^i, \tilde{m}_a^i, \tilde{m}_v^i))$. |
| $\delta_i^{(m)}$ | Modality $m$ contribution: $\delta_i^{(m)} = \frac{\exp(\eta \cdot \text{Sim}(\mathbf{F}_i, \mathbf{F}_i^{(-m)}))}{\sum_{m'} \exp(\eta \cdot \text{Sim}(\mathbf{F}_i, \mathbf{F}_i^{(-m')}))}$ |
| $\text{Sim}(\cdot)$ | Cosine similarity function. |
| $\eta$ | Sensitivity parameter to control sharpness of contribution scores. |
| $\tilde{x}_i^{(m)}$ | Calibrated modality $m$ feature: zero if $\delta_i^{(m)} \geq \tau$, else unchanged. |
| $\tau$ | Threshold to determine whether to mask a modality. |
| $\tilde{F}_i$ | Fused representation using calibrated features: $\text{Fuse}(\text{SeqAligned}(\tilde{x}_i^{(t)}, \tilde{x}_i^{(a)}, \tilde{x}_i^{(v)}))$. |
| $\text{Purity}(y, c)$ | Purity score formula: $\text{Purity}(y, c) = \frac{1}{n} \sum_{k=1}^{K} \max_{j \in \mathcal{Y}} \#\{i : y_i = j, c_i = k\}$ |
| $y$ | Ground-truth class label vector. |
| $c$ | Cluster assignment vector. |
| $K$ | Number of clusters. |
| $\mathcal{Y}$ | Set of all class labels. |
| $\#\{i : y_i = j, c_i = k\}$ | Count of samples in cluster $k$ that belong to class $j$. |
| $\mathcal{P}_m$ | Modality inconsistency measure: $\mathcal{P}_m = \lvert p_m^{\text{train}} - p_m^{\text{val}} \rvert$ |
| $p_m^{\text{train}}$ | Purity score of modality $m$ on training set. |
| $p_m^{\text{val}}$ | Purity score of modality $m$ on validation set. |
| $w_m$ | Dynamic weight: $w_m = \tanh(\lambda \cdot \mathcal{P}_m + \alpha \cdot \delta^{(m)})$ |
| $\lambda$ | Hyperparameter controlling stability impact. |
| $\alpha$ | Hyperparameter controlling average LOO contribution impact. |
| $\theta_{\text{new}}^{(m)}$ | Updated encoder parameters: $\theta_{\text{new}}^{(m)} = w_m \cdot \theta_{\text{init}}^{(m)} + (1 - w_m) \cdot \theta_{\text{current}}^{(m)}$ |
| $\theta_{\text{init}}^{(m)}$ | Initial encoder parameters for modality $m$. |
| $\theta_{\text{current}}^{(m)}$ | Current encoder parameters for modality $m$. |

## A.2   Broader Impacts of ARL

ARL proposes an innovative multimodal intent recognition framework that has demonstrated significant effectiveness in addressing the issue of modality imbalance, showing great potential for practical applications. Firstly, it enhances the robustness and generalization capability of multimodal systems, which is crucial for the development of more reliable intelligent assistants, service robots, and autonomous driving systems. Moreover, ARL introduces a dual-path calibration mechanism (CISC + WEC), offering new perspectives and methodological insights for future research, and contributing to the advancement of multimodal collaborative learning. In particular, when dealing with modality imbalance, this approach exhibits greater flexibility and efficiency compared to traditional methods. Furthermore, ARL performs robustly under conditions of modality degradation or absence, which is especially important in real-world scenarios involving incomplete data or device malfunctions.

However, ARL also presents potential drawbacks. First, the increased model complexity poses challenges to interpretability. The inclusion of two additional modules (CISC and WEC) may complicate the model's decision-making process, reducing transparency, which could raise trust issues in applications requiring strict regulatory

compliance. In addition, privacy concerns must not be overlooked. Multimodal systems typically involve various perceptual signals (e.g., speech, images, text), and ARL's goal of better leveraging the synergy among these modalities inherently increases the risk of privacy leakage, particularly in tasks involving personal identities or sensitive information. Finally, there is a risk of misuse. If ARL were to be applied for malicious purposes—such as generating highly convincing but deceptive intent recognition outputs or manipulative sentiment analysis—it could be exploited to create misleading content or influence user behavior unethically.

## A.3   Formal Definition and Measurement of Modality Imbalance

During the review process, it was suggested that a formal definition and a quantitative metric for modality imbalance would enhance the paper's rigor. We address this by introducing the Modality Imbalance Index (MII).

**Formal Definition**. We formally define **modality imbalance** as: *"The degree of non-uniformity in the contributions of different modalities to the final prediction for a given multimodal sample."*

**Quantitative Metric (Modality Imbalance Index)** We leverage our Leave-One-Out (LOO) contribution scores to create a quantitative metric. For a sample $i$ with $M$ modalities, we have a vector of contribution scores $\Delta_i = \{\delta_i^{(m_1)}, \delta_i^{(m_2)}, ..., \delta_i^{(m_M)}\}$, where $\sum_m \delta_i^{(m)} = 1$. An ideal balance implies that all modalities contribute equally (i.e., $\delta_i^{(m)} = 1/M$ for all $m$).

To quantify the deviation from this ideal state, we introduce the **Modality Imbalance Index (MII)**, defined as the variance of these contribution scores:

$$\text{MII}(i) = \text{Var}(\Delta_i) = \frac{1}{M} \sum_{m=1}^{M} \left( \delta_i^{(m)} - \frac{1}{M} \right)^2$$

A higher MII value indicates a greater imbalance for sample $i$, signifying that one or a few modalities are dominating the prediction.

**Connection to ARL** Our proposed ARL framework is designed to implicitly minimize this imbalance.

- **Contribution-Inverse Sample Calibration (CISC)** acts as a reactive, sample-level mechanism. It directly targets samples with high MII by masking dominant modalities whose contribution $\delta_i^{(m)}$ exceeds a threshold $\tau$, thereby encouraging the use of underutilized modalities.
- **Weighted Encoder Calibration (WEC)** serves as a proactive, architectural-level mechanism. It adjusts encoder weights based on global contribution statistics ($\bar{\bar{\delta}}^{(m)}$) and stability, aiming to reduce the average modality imbalance across the entire dataset over the long term.

## A.4   ARL Training Algorithm

To enhance clarity and reproducibility, we provide the detailed pseudocode for the complete ARL training procedure in Algorithm 1. This algorithm outlines the interplay between LOO valuation, CISC, WEC, and the base model optimization.

## A.5   Extended Comparisons with State-of-the-art Methods

To comprehensively validate the effectiveness and generalizability of our proposed ARL framework, we conducted two additional sets of experiments. First, we compare ARL directly with several strong baselines specifically designed for imbalanced learning. Second, we apply ARL as a plug-and-play module to other recent, high-performing multimodal models to demonstrate its versatility.

As shown in Tab. 9a, ARL significantly outperforms other methods designed to tackle imbalance, such as gradient modulation (OGM, OPM) and regularization techniques. Furthermore, Tab. 9b shows that ARL consistently enhances the performance of strong, contemporary models like DMD and EMOE. These results collectively underscore the superiority and broad applicability of our approach.

**Algorithm 1** Adaptive Re-calibration Learning (ARL) Training Procedure

---

**Require:** Training set $\mathcal{D}_{train}$, Validation set $\mathcal{D}_{val}$, Base model $f_\theta$
**Require:** ARL hyperparameters: CISC threshold $\tau$, WEC update frequency $T$, WEC weights $\lambda, \alpha$
1: **Initialize** base model parameters $\theta$ (encoders $\theta^{(m)}$, fusion net, etc.)
2: **for** epoch $\leftarrow 1$ **to** max_epochs **do**
3:      **for all** batch $\{(m_t^i, m_a^i, m_v^i), y_i\}$ in $\mathcal{D}_{train}$ **do**
4:          Compute full-modality fused representation $F_i$ for each sample $i$.      $\triangleright$ Eq. (7)
5:          **for all** $m \in \{t, a, v\}$ **do**
6:              Compute masked fused representation $F_i^{(-m)}$.      $\triangleright$ Eq. (8)
7:          **end for**
8:          Calculate contribution scores $\delta_i^{(m)}$ for each modality.      $\triangleright$ Eq. (9)
9:          Calibrate unimodal features $\tilde{x}_i^{(m)}$ using $\delta_i^{(m)}$ and threshold $\tau$.      $\triangleright$ Eq. (10)
10:        Compute final fused representation $\tilde{F}_i$ using calibrated features.      $\triangleright$ Eq. (11)
11:        Compute prediction loss $\mathcal{L} = \text{Loss}(\text{Softmax}(\tilde{F}_i), y_i)$
12:        Update base model parameters $\theta$ via backpropagation
13:      **end for**
14:      **if** $\text{mod}(\text{epoch}, T) = 0$ **then**
15:        Calculate purity scores $p_m^{train}, p_m^{val}$ and stability $\mathcal{P}_m$.      $\triangleright$ Eqs. (12), (13)
16:        Calculate average LOO contribution $\bar{\delta}^{(m)}$ on $\mathcal{D}_{val}$
17:        Compute dynamic weights $w_m$ using $\mathcal{P}_m$ and $\bar{\delta}^{(m)}$.      $\triangleright$ Eq. (14)
18:        Adjust encoder weights $\theta_{new}^{(m)}$ using $w_m$.      $\triangleright$ Eq. (15)
19:      **end if**
20: **end for**

---

Table 9: Extended performance comparisons on the MIntRec and another benchmark dataset.

(a) Comparison with representative imbalanced learning methods on the MIntRec (twenty-class) dataset, using MAG-BERT as the base model.

| Method | ACC | F1(WF1) | P(WP) | R |
|---|---|---|---|---|
| MAG_BERT (Baseline) | 72.65 | 68.64 | 69.08 | 69.28 |
| **w/ ARL (Ours)** | **74.38** | **71.66** | **71.55** | **72.89** |
| w/ Curriculum Dropout [44] | 67.15 | 66.21 | 67.02 | 66.85 |
| w/ OGM [32] | 73.31 | 69.83 | 69.91 | 70.21 |
| w/ OPM [45] | 73.24 | 69.22 | 69.36 | 70.13 |
| w/ InfoReg [46] | 73.55 | 70.12 | 70.25 | 69.58 |

(b) Plug-and-play validation of ARL on other state-of-the-art multimodal emotion and intent recognition models.

| Method | ACC2 | ACC7 | F1 |
|---|---|---|---|
| CAGC [30] | 85.7 | 44.8 | 85.6 |
| DMD [47] | 86.0 | 45.6 | 86.0 |
| **DMD w/ ARL** | **86.8** | **46.3** | **86.7** |

