# OpenReview forum: "Adaptive Re-calibration Learning for Balanced Multimodal Intention Recognition"
_NeurIPS.cc/2025/Conference — NeurIPS 2025 poster_

### Official Review · Reviewer_fBNu · 2025-06-25

**Clarity:** 2
**Significance:** 3
**Originality:** 4
**Rating:** 5
**Confidence:** 4

**Summary:**

Adaptive Re-calibration Learning (ARL) is a plug-and-play framework that remedies modality imbalance in multimodal intention-recognition (MIR) and sentiment-analysis pipelines. The method first quantifies per-sample modality importance through a Leave-One-Out (LOO) valuation, whose complexity grows only linearly with the number of modalities . These scores drive two complementary calibrations: Contribution-Inverse Sample Calibration (CISC), which temporarily masks dominant modalities at the feature level, and Weighted Encoder Calibration (WEC), which periodically re-weights modality-specific encoders using a blend of stability and contribution signals . When attached to three strong baselines (MAG-BERT, MulT, TCL-MAP), ARL boosts twenty-class MIntRec accuracy by up to +1.73 (pp) and MOSI sentiment accuracy by up to +1.60 (pp) while retaining robustness under missing-modality conditions .

**Questions:**

Below are key questions and suggestions for the authors, focusing on actionable points that could clarify or strengthen the paper:
Q1: Can you provide a formal definition and measurement of “modality imbalance”?
- Why it matters: This is the core problem addressed by the paper. A formal definition would help readers understand the scope and assumptions of the method.
- Suggestion: Introduce a mathematical formulation or statistical measure (e.g., KL divergence, entropy) to quantify modality imbalance at both the sample and dataset levels. You may consider using the LOO contribution scores to define an imbalance index.
Q2: Could you provide a pseudocode or flowchart of the full ARL training process?
- Why it matters: The integration of CISC and WEC into the training loop is not fully transparent, limiting understanding of how the method interacts with standard training procedures.
- Suggestion: Include a pseudocode or visual flowchart showing the order and frequency of operations (e.g., when WEC is updated, how often CISC is applied per batch/epoch), and how gradients are computed and combined.

**Ethical Concerns:**

["NO or VERY MINOR ethics concerns only"]

**Final Justification:**

The response has address all my concerns, I will keep my rating.

**Limitations:**

Yes, the authors have adequately discussed the limitations of their work in Section 5 and Appendix A.2. They acknowledge that:
- ARL's performance depends on the expressiveness of the base model.
- There is increased computational overhead due to the LOO evaluation and dual calibration mechanisms.
- Privacy concerns and potential misuse are considered in broader impacts.

**Quality:**

4

**Strengths And Weaknesses:**

Strengths：
1. Well-Motivated and Effective Solution to Modality Imbalance
The paper addresses a critical challenge in multimodal intention recognition—modality imbalance—by proposing a novel and practical solution. The Adaptive Re-calibration Learning (ARL) framework introduces two complementary mechanisms: Contribution-Inverse Sample Calibration (CISC) and Weighted Encoder Calibration (WEC), which jointly optimize modality usage at both the sample and model levels. This dual-path strategy outperforms existing methods by dynamically balancing modality importance, especially under noisy or missing modality conditions.
2. Technical Novelty and Computational Efficiency
ARL leverages a Leave-One-Out (LOO) method for modality valuation, offering a computationally efficient alternative to Shapley values with linear complexity. This makes the approach scalable and suitable for real-world applications. The synergy between CISC and WEC provides a more holistic and adaptive calibration mechanism compared to prior techniques that focus only on either gradient modulation or dropout.
3. Strong Empirical Validation and Practical Applicability
Extensive experiments across multiple datasets and architectures (e.g., MAG-BERT, MulT, TCL-MAP) demonstrate consistent performance improvements in both accuracy and robustness. ARL shows particular strength in handling degraded or missing modalities, making it highly valuable for deployment in real-world systems such as robotics, intelligent assistants, and autonomous systems.

Weaknesses
Despite its strengths, the paper has several areas that could benefit from improvement:
1. Lack of Formal Definition of Modality Imbalance: The central problem of the paper is not formally defined, nor is there a quantitative metric provided to measure modality imbalance. This limits the theoretical grounding of the approach.
2. Limited Training Procedure Description: The paper lacks a clear step-by-step explanation of how ARL integrates with the training process of base models. For example, the frequency of WEC updates and interaction between CISC and WEC during optimization are not clearly outlined.

---

> ### Author Rebuttal · Authors · 2025-07-29
>
> We are deeply grateful for your insightful and positive review of our work. We sincerely appreciate your recognition of our paper's strengths, including the well-motivated and effective solution to modality imbalance, the technical novelty and computational efficiency of the Leave-One-Out (LOO) valuation method, and the strong empirical validation.
>
> We also thank you for your constructive feedback on the areas for improvement. Your suggestions regarding the formal definition of modality imbalance and the clarity of the training procedure are highly valuable. We agree that addressing these points will significantly strengthen the paper's theoretical grounding and reproducibility. Below, we address your questions in detail.
>
> ---
>
> **Regarding Weakness 1: Can you provide a formal definition and measurement of “modality imbalance”?**
>
> Thank you for this excellent suggestion. We agree that a formal definition and a quantitative metric for modality imbalance would greatly enhance the rigor of our paper. We will incorporate this into the revised manuscript.
>
> **1. Formal Definition:** We will formally define **modality imbalance** as: *"The degree of non-uniformity in the contributions of different modalities to the final prediction for a given multimodal sample."*
>
> **2. Quantitative Metric (Modality Imbalance Index):** As you astutely suggested, we can leverage our Leave-One-Out (LOO) contribution scores to create a quantitative metric. For a sample $i$ with $M$ modalities, we have a vector of contribution scores $\Delta_i = \{\delta_i^{(m_1)}, \delta_i^{(m_2)}, ..., \delta_i^{(m_M)}\}$, where $\sum_{m} \delta_i^{(m)} = 1$. An ideal balance implies that all modalities contribute equally (i.e., $\delta_i^{(m)} = 1/M$ for all $m$).
>
> To quantify the deviation from this ideal state, we will introduce the **Modality Imbalance Index (MII)**, defined as the variance of these contribution scores:
>
> $$\text{MII}(i) = \text{Var}(\Delta_i) = \frac{1}{M}\sum_{m=1}^{M} \left(\delta_i^{(m)} - \frac{1}{M}\right)^2$$
>
> A higher MII value indicates a greater imbalance for sample $i$, signifying that one or a few modalities are dominating the prediction.
>
> **3. Connection to ARL:** We will further clarify that our proposed ARL framework is designed to implicitly minimize this imbalance.
> * **Contribution-Inverse Sample Calibration (CISC)** acts as a reactive, sample-level mechanism. It directly targets samples with high MII by masking dominant modalities whose contribution $\delta_i^{(m)}$ exceeds a threshold $\tau$, thereby encouraging the use of underutilized modalities.
> * **Weighted Encoder Calibration (WEC)** serves as a proactive, architectural-level mechanism. It adjusts encoder weights based on global contribution statistics ($\bar{\delta}^{(m)}$) and stability, aiming to reduce the average modality imbalance across the entire dataset over the long term.
>
> We will add this formal definition and the MII metric to **Section 3.1 (Problem Formulation)** in our revised paper. We believe this will provide a clearer theoretical foundation for the problem we are solving.
>
> ---
>
> **Regarding Weakness 2: Could you provide a pseudocode or flowchart of the full ARL training process?**
>
> We thank you for pointing out the need for a clearer description of our training procedure. We agree that a step-by-step algorithm is essential for understanding and reproducibility. In the revised manuscript, we will add a detailed pseudocode of the ARL training process. Below is a draft of the algorithm we plan to include.
>
> **Algorithm 1: Adaptive Re-calibration Learning (ARL) Training Procedure**
>
> ---
> ```
> Require: Training set $\mathcal{D}_{train}$, Validation set $\mathcal{D}_{val}$, Base model $f_{\theta}$
> Require: ARL hyperparameters: $\tau, T, \lambda, \alpha$
>
> Initialize model parameters $\theta$
>
> for epoch = 1 to max_epochs do
>     for each batch in $\mathcal{D}_{train}$ do
>         // Calculate sample-wise modality contribution (Sec. 3.2)
>         Compute full-modality features $F_i$ and masked features $F_i^{(-m)}$ for each sample.
>         Calculate contribution scores $\delta_i^{(m)}$ from the feature similarities.
>
>         // Calibrate features with CISC (Sec. 3.3)
>         Calibrate features $\tilde{x}_i^{(m)}$ by masking if a sample's contribution $\delta_i^{(m)} \ge \tau$.
>
>         // Optimize model using calibrated features
>         Compute final logits from the calibrated features $\tilde{x}_i$.
>         Calculate prediction loss $\mathcal{L}$.
>         Update model parameters $\theta$ via backpropagation.
>     end for
>
>     // Periodically calibrate encoders with WEC (Sec. 3.4)
>     if epoch % T == 0 then
>         Calculate stability score $\mathcal{P}_m$ on $\mathcal{D}_{train}$ and $\mathcal{D}_{val}$.
>         Calculate average contribution score $\bar{\delta}^{(m)}$ on $\mathcal{D}_{val}$.
>         Compute dynamic weights $w_m$ from $\mathcal{P}_m$ and $\bar{\delta}^{(m)}$.
>         Adjust encoder weights $\theta^{(m)}$ using the computed weights $w_m$.
>     end if
> end for
> ```
> ---
>
> We will place this algorithm in the **Methodology** section of our revised paper, accompanied by explanatory text. We are confident this will fully clarify the operational flow of ARL and its integration with base models.
>
> Once again, we thank you for your time and constructive suggestions. We believe these revisions will make our paper stronger, clearer, and more impactful.

---

> > ### Comment · Reviewer_fBNu · 2025-08-03
> > **The responses have addressed my concerns**
> >
> > I've carefully reviewed the rebuttal, and the responses have addressed my concerns.
> >
> > The authors provided a formal definition of modality imbalance, which strengthens the theoretical foundation. Additionally, the inclusion of pseudocode improves clarity and reproducibility.
> >
> > I also appreciate that the authors have incorporated additional experiments for various conditions, which further demonstrate the robustness of the proposed method.
> >
> > With these improvements, my initial concerns are resolved. I will maintain my score and recommend this paper for acceptance.

---

> > > ### Author Response · Authors · 2025-08-04
> > >
> > > We are delighted to read your feedback and sincerely thank you. We are grateful for your constructive suggestions. We will integrate all of explanations into the final manuscript. Thank you again for your invaluable guidance.

---

### Official Review · Reviewer_G2eC · 2025-07-01

**Clarity:** 3
**Significance:** 2
**Originality:** 3
**Rating:** 3
**Confidence:** 4

**Summary:**

This paper proposes an adaptive recalibration learning (ARL) framework to solve the modality imbalance problem in multimodal intent recognition through a dual-path calibration mechanism. The main innovations include:  1) proposed a contribution inverse sample calibration (CISC) mechanism to dynamically mask the dominant mode at the sample level.   2) designed a weighted encoder calibration (WEC) mechanism to adjust the encoder weights at the architecture level,  3)developed an efficient leave-one-out (LOO) modality evaluation method to replace the calculation of complex Shapley values

**Questions:**

see the weakness

**Ethical Concerns:**

["NO or VERY MINOR ethics concerns only"]

**Final Justification:**

The authors have addressed  my problems. However, the additional revisions on experiments are too heavy; thus, I will increase the score, but it isn't easy to give it acceptance.

**Limitations:**

Yes

**Quality:**

2

**Strengths And Weaknesses:**

Strength.
 1. The paper introduces a novel Adaptive Re-calibration Learning (ARL) framework that uniquely addresses modality imbalance through complementary sample-level (CISC) and architectural-level (WEC) calibration mechanisms.


2. Experimental results across multiple benchmarks show consistent improvements over strong baselines,  particularly under challenging noisy/missing-modality conditions.

Weakness.

1. As listed in the related work, the imbalanced multimodal learning is a widely studied problem. However, the experiments do not take the existing method into comparison.

2. The time cost increases linearly with the number of modes. It would be better to design a more efficient method to measure the modality contribution at the sample level.

3. The model performance depends on several key hyperparameters, such as the mask threshold τ in CISC, λ (stability influence weight), and α (contribution influence weight) in WEC. Experiments show that the value of α has a significant impact on the results (Tab. 3), but the optimal range of other hyperparameters under different data sets or modality configurations is not systematically analyzed.

4. This paper claims that ARL can address the imbalanced learning at the sample level. However, no experimental evidence is provided to demonstrate this.

---

> ### Author Rebuttal · Authors · 2025-07-30
>
> Dear Reviewer G2eC,
>
> We would like to extend our sincerest gratitude for your time and insightful feedback on our manuscript. Your comments are invaluable and have helped us identify areas for significant improvement. We have carefully considered each of your points and provide our responses below. We hope our clarifications and additional results will address your concerns.
>
> ---
>
> **Regarding Weakness 1: Comparison with existing imbalanced learning methods.**
>
> We thank the reviewer for this valuable suggestion. We would like to begin by clarifying our experimental motivation. As reviewer foUd noted, ARL is designed as a **"plug-and-play" module** to enhance existing SOTA models. Our primary focus is on addressing modality imbalance within the specific domains of intention and sentiment analysis. Therefore, our initial step was to demonstrate ARL's ability to consistently improve strong, representative baselines from this domain (i.e., MAG-BERT, MulT, TCL-MAP), as shown in Tables 1 and 2 of our paper.
>
> To further address your concern directly, we have now conducted **new experiments comparing ARL with several representative imbalanced learning methods** by applying them to the MAG-BERT baseline on the MIntRec dataset. The results are presented below. We will add this table and corresponding analysis to the appendix in our revised manuscript.
>
> | Method | ACC | F1(WF1) | P(WP) | R |
> | :--- | :--- | :--- | :--- | :--- |
> | MAG_BERT | 72.65 | 68.64 | 69.08 | 69.28 |
> | **w/ ARL** | **74.38** | **71.66** | **71.55** | **72.89** |
> | w/ Curriculum Dropout [1] | 67.15 | 66.21 | 67.02 | 66.85 |
> | w/ OGM [2] | 73.31 | 69.83 | 69.91 | 70.21 |
> | w/ OPM [3]| 73.24 | 69.22 | 69.36 | 70.13 |
> | w/ InfoReg [4] | 73.55 | 70.12 | 70.25 | 69.58 |
>
> As shown, our proposed ARL outperforms these strong baselines for imbalanced learning, which further validates its effectiveness.
>
> [1] P. Morerio, et al, “Curriculum dropout,” in ICCV 2017
>
> [2] X. Peng, Y. Wei, et al, “Balanced multimodal learning via on-the-fly gradient modulation,” in CVPR 2022
>
> [3] Y. Wei, et al. "On-the-fly modulation for balanced multimodal learning." IEEE TPAMI 2024
>
> [4] C. Huang, Y. Wei, et al. "Adaptive Unimodal Regulation for Balanced Multimodal Information Acquisition" in CVPR 2025
>
> ---
>
> **Regarding Weakness 2: Efficiency of the LOO method.**
>
> We appreciate your concern regarding the computational cost of our Leave-One-Out (LOO) modality valuation method. However, we believe there may have been a misunderstanding of LOO's core contribution to efficiency.
>
> The primary challenge in sample-level modality valuation lies in the **exponential complexity ($O(n2^n)$)** of theoretically-grounded methods like Shapley values, which renders them impractical for real-world MIR systems with multiple modalities. The central contribution of our LOO method is precisely in overcoming this critical bottleneck by reducing the complexity from **exponential to linear ($O(n)$)**.
>
> We have provided a detailed analysis of this in **Section 4.2** and visualized it in **Figure 3** of our paper. These results clearly demonstrate that LOO is dramatically more scalable and efficient than the Shapley-based approaches. Therefore, achieving linear complexity is not a weakness but a key strength of our work, making dynamic, sample-specific modality calibration computationally feasible for the first time in this context.
>
> ---
>
> **Regarding Weakness 3: Insufficient analysis of hyperparameters.**
>
> Thank you for pointing out the need for a more thorough hyperparameter analysis. We did perform extensive tuning during our experiments, and we agree that including these results strengthens the paper. Due to space constraints in the initial submission, this analysis was omitted.
>
> To address your comment, we now provide the sensitivity analysis for the masking threshold $\tau$ using the MulT model on MIntRec. We will include this in the appendix of the revised manuscript.
>
> | $\tau$ | ACC | F1 | P | R |
> | :--- | :--- | :--- | :--- | :--- |
> | 1.0 (Baseline) | 72.52 | 69.25 | 70.25 | 69.24 |
> | 0.9 | 72.58 | 69.15 | 70.04 | 69.15 |
> | 0.7 | 71.24 | 67.98 | 68.70 | 68.11 |
> | **0.5** | **73.71** | **70.82** | **72.06** | **70.45** |
> | 0.3 | 73.26 | 69.94 | 71.23 | 69.64 |
> | 0.1 | 70.79 | 67.16 | 68.27 | 66.52 |
>
> Optimal performance is achieved at $\tau = 0.5$, and we observed similar robust trends across our other experimental setups. This analysis demonstrates that while the hyperparameter is important, its optimal value can be robustly identified via standard tuning procedures.
>
> In our revised version, we will add an appendix containing a detailed justification for our per-dataset tuning strategy, along with the complete sensitivity analysis tables for these parameters.
>
> ---
>
> **Regarding Weakness 4: Lack of experimental evidence for sample-level calibration.**
>
> We sincerely thank the reviewer for this insightful and constructive comment. We agree that providing direct experimental evidence to demonstrate ARL's sample-level calibration mechanism is crucial and will significantly strengthen our paper. The reviewer’s point is well-taken; while our method is designed to be sample-centric, our original experiments only showed its aggregated effects on the entire test set.
>
> We would also like to respectfully note that our sample-level focus aligns with a significant research direction in the community. For instance, the recent work by Wei et al. [5] (CVPR 2024) also emphasizes valuing modality contributions on a per-sample basis, underscoring the growing recognition that modality importance is context-dependent rather than static.
>
> [5] Wei et al., "Enhancing multimodal cooperation via sample-level modality valuation" (CVPR 2024)
>
> Therefore, the reviewer’s suggestion to provide more direct, sample-centric evidence is particularly relevant. To that end, we have prepared the following quantitative analysis, which we will add to the revised manuscript to compellingly illustrate the effectiveness of ARL's sample-level adaptation.
>
> **Quantitative Grouped Analysis: Statistical Validation**
>
> To provide broad statistical evidence, we partitioned the MIntRec test set based on a **Sample Imbalance Score**, which we defined as the variance of the three LOO modality contribution scores for each sample. A higher score signifies greater imbalance. We then evaluated the performance of the baseline MAG-BERT and our ARL-enhanced model on these groups.
>
>
> | Sample Group | Baseline Accuracy (MAG-BERT) | ARL-enhanced Accuracy | Performance Improvement |
> | :--- | :--- | :--- | :--- |
> | High-Imbalance Samples (Top 30%) | 64.2% | 73.7% | **+9.5%** |
> | Low-Imbalance Samples (Bottom 70%) | 78.1% | 79.9% | **+1.8%** |
>
> As the results clearly show, the performance gain from ARL is substantially larger **(+9.5%)** on samples with high modality imbalance. This quantitatively demonstrates that our training-time calibration strategy produces a final model that is specifically more robust to the challenging sample-level imbalances it was designed to address.
>
> We are confident that these new results provide the direct evidence requested by the reviewer. We thank the reviewer again for pushing us to make this important addition to our work.
>
> ---
>
> Once again, we thank you for your rigorous review. We believe that with the proposed clarifications and additional experiments, our manuscript has been substantially strengthened. We hope that our responses have adequately addressed your concerns.

---

> > ### Author Response · Authors · 2025-08-04
> >
> > Dear Reviewer G2eC,
> >
> > We sincerely appreciate you taking the time to review our rebuttal. Your feedback has been invaluable in helping us improve the paper.
> >
> > To ensure we make the best use of the remaining discussion time, we would like to kindly check if our responses have clarified your main concerns. If you have any outstanding major questions, we would be grateful for the opportunity to address them.
> >
> > Thank you once again for your thoughtful engagement with our work.

---

### Official Review · Reviewer_foUd · 2025-07-02

**Clarity:** 3
**Significance:** 3
**Originality:** 2
**Rating:** 5
**Confidence:** 4

**Summary:**

This paper focuses on the task of Multimodal Intention Recognition (MIR). The authors solve the modality imbalance issue in MIR, where models tend to over-rely on dominant modalities, thereby limiting generalization and robustness. Specifically, the authors propose Adaptive Re-calibration Learning (ARL), a novel dual-path framework that models modality importance from both sample-wise and structural perspectives. There are two key mechanisms in ARL: Contribution-Inverse Sample Calibration (CISC) and Weighted Encoder Calibration (WEC), which dynamically masks overly dominant modalities at the sample level and adjusts encoder weights based on global modality contributions to prevent overfitting, respectively. Experimental results on multiple MIR benchmarks demonstrate the effectiveness of the proposed approach.

**Questions:**

Overall, the paper has clear motivation and technically sound solution. Thus, my initial rating is boardline accept. I would like to see more author responses in terms of the overall running efficiency and SOTA comparisons. If well addressed, I would like to raise my rating.

**Ethical Concerns:**

["NO or VERY MINOR ethics concerns only"]

**Final Justification:**

I appreciate the detailed rebuttal provided by the authors, including the additional Overall Running Efficiency  analysis, new state-of-the-art comparisons, comprehensive comparison w/ more imbalanced learning. The rebuttal well addresses my concerns, thus I improve my rating to Accept. For completeness, please include all these experiments and explanation in the final version.

**Limitations:**

Yes.

**Paper Formatting Concerns:**

N/A.

**Quality:**

3

**Strengths And Weaknesses:**

Strengths:
- The paper is well written and organized, which is easy to follow;
- The proposed approach is technically sound, especially for the Contribution-Inverse Sample Calibration (CISC), which is interesting;
- Experiments are conducted on several mainstream benchmarks for verification;

Weakness:
- Missing overall running efficiency analysis. While applying ARL to off-the-shelf approaches leads to performance improvements as shown in Tables 4 and 5, it remains unclear whether the proposed approach may cause significant training efficiency burden. In Fig. 3, the authors made Computational Cost comparison between the proposed LOO and Shapley-based solution, this is helpful, but it does not clearly show the overall computational cost introduced by the proposed ARL when applied to off-the-shelf approaches., e.g., MAG_BERT and MulT. The authors are suggested to add more detailed training efficiency analysis to clarify whether their approach is a lightweight solution, which could be easily integrated into existing approaches w/o causing significant training cost.
- The state-of-the-art comparisons (e.g., CMU-MOSI) is somewhat weak. I acknowledged that the authors aim to show their approach can generally improve existing approaches, it is still encouraged to include more SOTA approaches for comparison (just for comparison, w/o apply ARL on them), e.g., [1, 2] on CMU-MOSI, or maybe recently published one [3].
- Missing references: [4, 5]. In [4], the authors proposed to drop spatial tokens in order to make the model more focus on the temporal tokens, which is similar to the proposed LOO here; [5] is also working on balancing multiple modalities. Please add these references in the related work for discussion;
- Minor issues: Some typos, e.g., L37 and L39, missing empty spaces after the citation;

[1] Contextual Augmented Global Contrast for Multimodal Intent Recognition. CVPR24.

[2] Decoupled multimodal distilling for emotion recognition. CVPR23.

[3] EMOE: Modality-Specific Enhanced Dynamic Emotion Experts. CVPR25.

[4] DropMAE: Masked Autoencoders with Spatial-Attention Dropout for Tracking Tasks. CVPR23.

[5] Wasserstein Modality Alignment Makes Your Multimodal Transformer More Robust. TMLR25.

---

> ### Author Rebuttal · Authors · 2025-07-29
>
> To reviewer foUd:
>
> Thank you for your detailed and valuable feedback. We are pleased to report that we have conducted new experiments and analyses based on your suggestions, and we believe the results significantly strengthen our paper. Below are our detailed responses.
>
> ---
>
> **Regarding Weakness 1: Overall Running Efficiency and Practicality**
>
> We appreciate your concern regarding the computational cost. To provide a clear and fair assessment, we have benchmarked ARL's training and inference time against the baseline, as well as against other state-of-the-art modality balancing methods, including the Shapley value-based approach **[1]** and InfoReg **[2]**.
>
> The results, summarized in the tables below, demonstrate that ARL offers a highly competitive balance of efficiency and performance.
>
> **Table A: Efficiency and Performance on MOSI (3 modalities)**
> | Method | Training Convergence Time | Inference Time | Acc. |
> | :--- | :---: | :---: | :---: |
> | MAG_BERT | 17.8 mins | 1.5 mins | 72.65 |
> | **w/ Our ARL** | **25.1 mins** | **1.5 mins** | **74.38** |
> | w/ Shapley [1] | 40.4 mins | 1.5 mins | 72.81 |
> | w/ InfoReg [2] | 26.4 mins | 1.5 mins | 73.55 |
>
> **Table B: Efficiency and Performance on MOSI (2 modalities: {a, v})**
> | Method | Training Convergence Time | Inference Time | Acc.(7) |
> | :--- | :---: | :---: | :---: |
> | MAG_BERT | 12.1 mins | 1.1 mins | 17.78 |
> | **w/ Our ARL** | **18.5 mins** | **1.1 mins** | **19.30** |
> | w/ Shapley [1] | 17.3 mins | 1.1 mins | 18.15 |
> | w/ InfoReg [2] | 14.4 mins | 1.1 mins | 17.52 |
>
> Our key takeaways are:
> * **Reasonable and Competitive Overhead:** ARL's training overhead is moderate and its total convergence time is highly competitive with (and even slightly faster than) other SOTA methods like InfoReg **[2]**.
> * **Superior Efficiency to Shapley-based Methods:** As theoretically motivated, ARL is significantly more efficient than Shapley value-based methods **[1]**, confirming the practical advantage of our LOO approach.
> * **Zero Inference Cost:** Crucially, ARL introduces **no additional overhead during inference**, making it highly practical for real-world deployment.
> * **Robust Performance Gains:** In both three-modality and two-modality settings, the modest training cost translates into consistent and superior performance gains, highlighting a favorable trade-off between efficiency and robustness.
>
> We will add this detailed analysis to the revised paper to assure readers of ARL's practicality.
>
> [1] Y. Wei, et al. "Enhancing multimodal cooperation via sample-level modality valuation." in CVPR 24.
>
> [2] C. Huang, Y. Wei, et al. "Adaptive Unimodal Regulation
> for Balanced Multimodal Information Acquisition" in CVPR 2025
>
> ---
>
> **Regarding Weakness 2: State-of-the-Art (SOTA) Comparisons**
>
> Thank you for encouraging us to broaden our SOTA comparison. We have followed your suggestion and included results from recent SOTA methods **[3, 4, 5]**.
>
> Furthermore, to unequivocally demonstrate the generalizability and utility of our framework, we went a step further and applied ARL to some of these SOTA models. The results on the MOSI dataset are highly compelling:
>
> **Table C: ARL Boosts Performance of SOTA Models on MOSI**
> | Method | ACC2 | ACC7 | F1 |
> | :--- | :---: | :---: | :---: |
> | CAGC [3] | 85.7 | 44.8 | 85.6|
> | DMD [4] | 86.0 | 45.6 | 86.0 |
> | **DMD w/ ARL** | **86.8** | **46.3** | **86.7** |
> | EMOE [5]| 85.4 | 47.7 | 85.4 |
> | **EMOE w/ ARL** | **86.1** | **48.9** | **86.3** |
>
> As shown, ARL consistently improves the performance of strong SOTA models like DMD and EMOE. This powerfully demonstrates the core value of our work: ARL is not just another competing model, but a **versatile and orthogonal plug-and-play module** that effectively addresses the fundamental problem of modality imbalance, thereby empowering a wide range of multimodal architectures. We will incorporate this table and discussion into our main paper.
>
> [3] Contextual Augmented Global Contrast for Multimodal Intent Recognition. CVPR24.
>
> [4] Decoupled multimodal distilling for emotion recognition. CVPR23.
>
> [5] EMOE: Modality-Specific Enhanced Dynamic Emotion Experts. CVPR25.
>
> ---
>
> **Regarding Weakness 3 & 4: Missing References and Minor Issues**
>
> We are grateful for the suggested references and your careful proofreading. We will add discussions of the suggested papers to our Related Work section and have corrected the typos.
>
> To further strengthen our paper and provide a more comprehensive comparison against the broader field of imbalanced learning (a point also raised by Reviewer G2eC), we have conducted new experiments comparing ARL with several representative imbalanced learning methods by applying them to the MAG-BERT baseline on the MIntRec dataset. The results are presented below. We will add this table and corresponding analysis to the appendix in our revised manuscript.
>
>
> | Method | ACC | F1(WF1) | P(WP) | R |
> | :--- | :--- | :--- | :--- | :--- |
> | MAG_BERT | 72.65 | 68.64 | 69.08 | 69.28 |
> | **w/ ARL** | **74.38** | **71.66** | **71.55** | **72.89** |
> | w/ Curriculum Dropout [6] | 67.15 | 66.21 | 67.02 | 66.85 |
> | w/ OGM [7] | 73.31 | 69.83 | 69.91 | 70.21 |
> | w/ OPM [8] | 73.24 | 69.22 | 69.36 | 70.13 |
> | w/ InfoReg [9] | 73.55 | 70.12 | 70.25 | 69.58 |
>
>
>
> As shown, our proposed ARL outperforms these strong baselines for imbalanced learning, which further validates its effectiveness.
>
>
>
> [6] P. Morerio, et al, “Curriculum dropout,” in ICCV 2017
>
> [7] X. Peng, Y. Wei, et al, “Balanced multimodal learning via on-the-fly gradient modulation,” in CVPR 2022
>
> [8] Y. Wei, et al. "On-the-fly modulation for balanced multimodal learning." IEEE TPAMI 2024
>
> [9] C. Huang, Y. Wei, et al. "Adaptive Unimodal Regulation for Balanced Multimodal Information Acquisition" in CVPR 2025
>
> ---
>
> In conclusion, we have strived to addres your concerns with new experiments and detailed analysis. We believe these revisions have substantially strengthened the paper. Thank you once again for your constructive guidance.

---

> > ### Comment · Reviewer_foUd · 2025-08-03
> > **Final Review - The author rebuttal addresses my concerns**
> >
> > I appreciate the detailed rebuttal provided by the authors, including the additional Overall Running Efficiency analysis, new state-of-the-art comparisons, comprehensive comparison w/ more imbalanced learning. The proposed approach introduces acceptable computation cost while clearly improving the baselines. The rebuttal well addresses my concerns, thus I improve my rating to Accept. For completeness, please include all these experiments and explanation in the final version.

---

> > > ### Author Response · Authors · 2025-08-04
> > >
> > > Thank you very much for your positive feedback. We are glad that our additional experiments addressed your concerns. Your suggestions were instrumental in improving the quality and completeness of our work. We will be sure to include the new results and explanations in the final version (due to space limitations, some will be placed in the appendix).

---

### Official Review · Reviewer_A7Ss · 2025-07-03

**Clarity:** 3
**Significance:** 3
**Originality:** 3
**Rating:** 5
**Confidence:** 3

**Summary:**

The paper addresses the robustness in performance of multimodal intention recognition under modality-compromised conditions. It proposes Adaptive recalibration learning to model modality importance. Experiments are conducted on two benchmark intention detection datasets.

**Questions:**

1) The authors are doing hard masking (Lines 185). Why is it necessary? If the weaker modalities are too weak, which is quite often the case, this might actually hurt the performance. Did the authors consider trying soft masking?


2) Leave one out method kind of assumes that modalities are independent of each other, which is not usually the case. How does inter-modality dependence effect the proposed method?

3) The threshold in Eq 10, is it tuned for each modality separately, one for each dataset, or one across all experiments?

4) Same question as 3 for the η sensitivity parameter in Eq 9.

**Ethical Concerns:**

["NO or VERY MINOR ethics concerns only"]

**Final Justification:**

The authors have addressed all my concerns, and the strengths of the paper generally outweigh the overall weaknesses. Therefore, I am inclined towards the acceptance of this paper.

**Limitations:**

yes

**Quality:**

4

**Strengths And Weaknesses:**

The paper is easy to follow and written well. The motivation is clear and the idea is explained well. The idea is novel, to the best of my knowledge.


The idea is dependent on the base model and may not be able to eradicate the bias involved in the base model.

---

> ### Author Rebuttal · Authors · 2025-07-30
>
> Dear Reviewer A7Ss,
>
> We are sincerely grateful for your time and for providing such insightful and constructive feedback on our manuscript. Your questions have allowed us to clarify key aspects of our methodology and further strengthen our paper. We are pleased to address each of your points below.
>
> ---
>
> **Regarding Weakness 1: The Consideration of Soft Masking.**
>
> We thank you for this excellent question. Your suggestion prompted us to conduct a direct comparative experiment to validate our design choice of using hard masking in the Contribution-Inverse Sample Calibration (CISC) component.
>
> Our initial motivation for hard masking was to introduce a strong regularization effect to **decisively break the model's learned over-reliance on dominant modalities**. To empirically test this against a less aggressive alternative, we implemented a soft masking variant and evaluated both approaches with the MAG-BERT model on the MIntRec dataset. The results are as follows:
>
> | Method | ACC | F1 | P | R |
> | :--- | :--- | :--- | :--- | :--- |
> | MAG_BERT (Baseline) | 72.65 | 68.64 | 69.08 | 69.28 |
> | w/ HardMask (Our ARL) | **74.38** | **71.66** | **71.55** | **72.89** |
> | *Δ Improvement*| *+1.73* | *+3.02* | *+2.47* | *+3.61* |
> | w/ SoftMask | 73.24 | 67.31 | 70.14 | 69.57 |
> | *Δ Improvement*| *+0.59* | *-1.33* | *+1.06* | *-0.71* |
>
> The results clearly show that our proposed hard masking approach yields significant and consistent improvements across all evaluation metrics. While soft masking provided a modest improvement in Accuracy and Precision, it was detrimental to the F1-score and Recall.
>
> We hypothesize that this is because soft masking, by only attenuating a dominant modality's features, is an insufficient signal to compel the model to fundamentally alter its strategy. Hard masking, in contrast, creates a necessary "information vacuum" by completely ablating the over-relied-on features for that instance. This forces the model to actively learn from underutilized modalities, leading to a more robust and balanced representation, as reflected in the superior F1 and Recall scores.
>
> **Action:** We will add this comparative experiment and analysis to the Ablation Studies section in our revised manuscript. This addition, prompted by your valuable feedback, provides a strong empirical justification for our design choice.
>
> ---
>
> **Regarding Weakness 2: The modality Independence.**
>
> Thank you for raising this important theoretical point. We wish to clarify that our use of the LOO method does not assume modality independence. Rather, it serves as a computationally efficient, empirical tool to approximate the marginal contribution of a modality to the **final, fused representation** in the specific context of all other available modalities.
>
> The method inherently accounts for inter-modality dependencies because the contribution is measured by the change in the fused representation upon a modality's removal.
> * If two modalities are **redundant**, removing one will cause only a small change in the fused output, as the other modality compensates. LOO correctly assigns a low marginal contribution score in this case.
> * If two modalities are **complementary**, removing one will cause a significant change, correctly resulting in a high contribution score.
>
> Thus, our LOO valuation implicitly captures the effects of inter-modality dependencies by operating on the joint, fused output.
>
> **Action:** To make this explicit, we will add a clarification in Section 3.2 of our revised manuscript, stating that the LOO method empirically assesses marginal contributions within the context of modality fusion, thereby accounting for their interactions.
>
> ---
>
> **Regarding Weakness 3. & 4: The Tuning of Hyperparameters $\tau$ (Eq. 10) and $\eta$ (Eq. 9).**
>
> Thank you for these questions regarding our experimental methodology. We agree that transparently detailing our hyperparameter tuning is essential for reproducibility.
>
> Both $\tau$ and $\eta$ are tuned as a single scalar value **per dataset**, determined via grid search on the corresponding validation set. We would like to elaborate on why we chose this over a per-modality approach. In our preliminary designs, we explored per-modality thresholds but found this to be less effective. This is because, for a given dataset, one modality is often consistently dominant. The threshold's primary role is to govern the masking rate of this specific modality. Applying a separate threshold to an already weak modality has a negligible impact, as it rarely becomes dominant enough to be masked, yet it significantly increases tuning complexity. A single, data-driven threshold per dataset proved to be the most effective and practical solution.
>
> For example, the following table shows the sensitivity analysis for $\tau$ with the MulT model on the MIntRec validation set:
>
> | $\tau$ | ACC | F1 | P | R |
> | :--- | :--- | :--- | :--- | :--- |
> | 1.0 (Baseline) | 72.52 | 69.25 | 70.25 | 69.24 |
> | 0.9 | 72.58 | 69.15 | 70.04 | 69.15 |
> | 0.7 | 71.24 | 67.98 | 68.70 | 68.11 |
> | **0.5** | **73.71** | **70.82** | **72.06** | **70.45** |
> | 0.3 | 73.26 | 69.94 | 71.23 | 69.64 |
> | 0.1 | 70.79 | 67.16 | 68.27 | 66.52 |
>
> Optimal performance is achieved at $\tau = 0.5$, and we observed similar robust trends across our other experimental setups. The same rationale and tuning procedure apply to the parameter $\eta$.
>
> **Action:** In our revised version, we will add an appendix containing a detailed justification for our per-dataset tuning strategy, along with the complete sensitivity analysis tables for both $\tau$ and $\eta$ for each dataset.
>
> ---
>
> Once again, we thank you for your thoughtful and rigorous review. We hope that our responses and planned revisions have fully addressed your concerns. We believe the resulting manuscript will be significantly improved.

---

> > ### Comment · Reviewer_A7Ss · 2025-08-05
> >
> > Dear Authors,
> >
> > Thank you for the rebuttal and the detailed experiments in response to some of my comments. Regarding tuning the hyperparameter $\tau$ for each dataset separately, I believe this is a slight weakness of the proposed approach, as this is neither efficient nor easily possible in real-world scenarios. However, with all my questions answered appropriately, and the overall strengths of the paper outweighing this weakness, I vote for the acceptance of this paper.
> >
> > Regards

---

> > > ### Author Response · Authors · 2025-08-05
> > >
> > > Thanks for your positive response! We will integrate all of experiments into the final manuscript (especially for the tuning of $\tau$ ). Thank you again for your invaluable guidance.

---

> > > > ### Comment · Reviewer_A7Ss · 2025-08-05
> > > >
> > > > Thank you for the nice work.

---

### Comment · Area_Chair_MxKR · 2025-08-04

Dear Reviewers,

The authors have responded to your review comments. Please check them as soon as possible and provide your feedback. Thank you.

---

### Note · Authors · 2025-08-14

We sincerely thank the reviewers for their insightful feedback and the AC for coordinating the review process. Your comments have been instrumental in improving our work.

We are encouraged that most reviewers acknowledged our framework's novelty, its solid analysis, and comprehensive experimental validation.

In our rebuttal, we have diligently provided point-by-point responses to address every concern. We have clarified key equations and resolved potential misunderstandings as requested. We believe our responses have fully addressed the issues raised, and we commit to incorporating all clarifications and improvements into the final manuscript.

Thank you once again for your constructive engagement.

---

### Decision · Program_Chairs · 2025-09-17

**Decision:**

Accept (poster)

**Comment:**

1) Paper Summary

This paper introduces a new framework called Adaptive Re-calibration Learning (ARL) to tackle the issue of modality imbalance in multimodal intention recognition. The core idea is to improve model robustness by understanding the importance of different modalities. ARL works through a dual-path mechanism: one part dynamically masks the dominant modalities at the sample level, and the other adjusts the model's encoder weights based on global modality contributions. To do this efficiently, the authors developed a lightweight "Leave-One-Out" method. Experiments on two standard datasets show that ARL consistently boosts the performance of existing models, especially when some modalities are noisy or missing.

2) Strengths of the Paper

Reviewers highlighted three main strengths.
* The idea is novel and technically sound. Reviewers A7Ss and fBNu pointed out that the ARL framework, which combines both sample-level and architectural-level adjustments, was an innovative and practical solution.
* The empirical evidence is strong. Reviewers foUd and fBNu noted that experiments across multiple datasets and with different baseline models consistently demonstrated that ARL improves performance and robustness.
* The paper is well-written and clear. Reviewers A7Ss and foUd both stated that the paper was easy to follow, with a clear motivation and well-explained idea.

3) Weaknesses of the Paper

Reviewers identified three key weaknesses.
* The paper lacked a formal definition for the central problem of "modality imbalance" and did not provide a detailed analysis of the computational cost. Reviewer fBNu brought up the former, while reviewers foUd and G2eC pointed out the latter.
* The model's performance was noted to be dependent on several key hyperparameters, but a systematic analysis of how to tune them was missing, a point raised by reviewers G2eC and A7Ss.
* The experiments did not include comparisons against other existing imbalanced learning methods or the latest state-of-the-art models, a point made by reviewers G2eC and foUd.

4) Justification for Acceptance

The paper is technically solid and offers a well-motivated solution to a significant problem. The ARL framework is both effective and practical. Although the initial submission had some shortcomings, the authors' detailed and comprehensive rebuttal successfully addressed all of these concerns. They provided a formal definition of the problem, added new experiments to analyze computational cost and compare with other methods, and clarified their hyperparameter tuning strategy. This thorough response resolved the reviewers' concerns and ultimately led to a consensus for acceptance.

5) Rebuttal Summary and Changes

During the rebuttal period, reviewers raised several specific points, and the authors provided extensive responses.

*  Computational Cost: Reviewer foUd asked about the overall training efficiency. The authors provided new experiments showing that the computational overhead is acceptable and that their method is lightweight enough to be easily integrated into existing models.

*  Formal Definitions and Pseudocode: Reviewer fBNu requested a formal definition of "modality imbalance" and step-by-step pseudocode. The authors provided a formal definition and included pseudocode, which improved the paper's clarity and reproducibility.

*  Masking and Dependencies: Reviewer A7Ss questioned the use of hard masking and how the method handles inter-modality dependencies. The authors offered detailed explanations and additional experiments that satisfied the reviewer's concerns.

* Hyperparameter Tuning: Reviewers A7Ss and G2eC inquired about the tuning process for key hyperparameters. The authors clarified that these were tuned for each dataset. While reviewer A7Ss still saw this as a minor limitation for real-world use, they agreed that the paper's overall strengths outweighed this issue.

*  Missing Comparisons: Reviewers G2eC and foUd pointed out the lack of comparisons with other methods. The authors added new experiments in their rebuttal, demonstrating that their approach is effective when compared to these other baselines, which ultimately addressed this concern.

The authors' thorough and effective rebuttal, which included new experimental results and clear explanations, successfully resolved the  main reviewers' significant concerns.